

# Concurrent anemia and stunting among schoolchildren in Wonago district in southern Ethiopia: a cross-sectional multilevel analysis

Hiwot Hailu Amare[1,2,3] and Bernt Lindtjorn[1,2]

[1] School of Public Health, College of Medicine and Health Sciences, Hawassa University, Hawassa, Ethiopia
[2] Centre for International Health, University of Bergen, Bergen, Norway
[3] Department of Public Health, College of Health Sciences and Medicine, Dilla University, Dilla, Ethiopia

Corresponding author
Hiwot Hailu Amare,
hiwothailu14@yahoo.com

## ABSTRACT

**Background**. Even if many schoolchildren in Ethiopia are anemic and stunted, few have studied the co-existence of anemia and stunting among schoolchildren in Ethiopia. In addition, multilevel analysis to explore the variation in prevalence of concurrent anemia and stunting (CAS) across schools and classes is rarely applied. Thus, we aimed to assess the prevalence and risk factors of CAS at the individual, household, and school level among schoolchildren in southern Ethiopia.

**Methods**. We recruited 864 students aged 7–14 years from the Wonago district in southern Ethiopia using a three-stage random sampling, assigning four schools to level one, 24 classes to level two. We then randomly selected 36 children from each class, and recorded their weight, height, haemoglobin, intestinal helminthic infections, hygienic practices, dietary practices, household food insecurity, and socio-demographic information. A multivariate, multilevel logistic regression model was applied to detect potential risk factors for CAS.

**Results**. The prevalence of CAS was 10.5% (85/810) among schoolchildren, which increased with age in years (adjusted odds ratio [aOR] 1.39 [95% confidence interval 1.13, 1.71, $P = 0.002$]) and among children who always did not wash their hands with soap after use of latrine (aOR 4.30 [1.21, 15.3, $P = 0.02$]). Children who walked barefoot (aOR 10.4 [2.77, 39.1, $P = 0.001$]), were infected with *Trichuris trichiura* (aOR 1.74 [1.05, 2.88, $P = 0.03$]), or had head lice infestation (aOR 1.71 [1.01, 2.92, $P = 0.04$]) had higher CAS prevalence. Prevalence rates of CAS were low in those using treated drinking water (aOR 0.32 [95% CI 0.11, 0.97, $P = 0.04$]). Most of the risk factors for CAS were identified at the individual level. The clustering effect measured by the intra-cluster correlation coefficient was 6.8% at school level and 19% at class.

**Conclusion**. CAS prevalence is a moderate public health problem among schoolchildren in southern Ethiopia and varies across classes and schools. After controlling for clustering effects at the school and class levels, we found an association between CAS and increasing age, not always washing hands with soap after using latrine, walking barefoot, and *T. trichiura* infection. Using treated water for drinking was found to have a protective effect against CAS. Thus, educating children on personal hygiene and provision of safe drinking water could reduce the CAS burden in schoolchildren in rural areas of southern Ethiopia.

## INTRODUCTION

Nearly half of all countries worldwide face the multiple burdens of malnutrition, including stunting, wasting, poor weight gain, and micronutrient deficiencies (*International Food Policy Research Institute, 2015*). Ethiopia has a school nutrition program designed to improve students' nutritional status (*Federal Democratic Republic of Ethiopia Ministry of Education, 2012*; *Federal Democratic Republic of Ethiopia Ministry of Health, 2017*). Despite school health and nutrition initiatives, a large proportion of schoolchildren in Ethiopia are still anemic or stunted (*Federal Democratic Republic of Ethiopia Ministry of Education, 2012*; *Federal Democratic Republic of Ethiopia Ministry of Health, 2017*). In Ethiopia, even though school access and enrolment have improved, about 10 million school-age children live in food-insecure areas (*Federal Ministry of Education, 2015*). Southern Ethiopia is among the regions in the country with inadequate school water supply (*Federal Democratic Republic of Ethiopia Ministry of Education, 2017*), and with poor sanitation practice with more than half of the schools still using open defecation (*Grimes et al., 2017*).

Anemia affects about one-third of the population worldwide. Although the prevalence varies across different population groups and settings, the most common cause is iron deficiency (*Kassebaum et al., 2014*). Although there are few recent global reports for anemia among school-aged children, WHO estimated in 2008 a 25% prevalence among school-aged children (*De Benoist et al., 2008*). In 2016, the national prevalence of anemia among schoolchildren was 26% in Ethiopia, 20% with iron deficiency anemia (*Ethiopian Public Health Institute, 2016*) and similar results are found in southern Ethiopia (*Grimes et al., 2017*; *Shaka & Wondimagegne, 2018*). Stunting is also common among schoolchildren in southern Ethiopia with prevalence rates ranging from 15% to 57% (*Bogale et al., 2018*; *Shaka & Wondimagegne, 2018*). Both anemia and stunting are preventable or treatable (*Kakietek et al., 2017*).

Studies indicate a link between anemia and stunting (*Getaneh et al., 2017*; *Gutema et al., 2014*). Stunting is often associated with anemia and recurrent infections among children (*Gupta, 2017*). Children with poor nutrient intake have weakened immunity, are susceptible to infections, and are at increased risk of anemia and long-term developmental delays (*Bourke, Berkley & Prendergast, 2016*; *Chaparro & Suchdev, 2019*). Despite these complex and interconnected health problems, few studies have investigated the co-existence of anemia and stunting in Ethiopia. Apart from the 2016 Ethiopian Demographic and Health Survey which demonstrated a prevalence rate of 24% of the co-morbidity of anemia and stunting among children aged under-5 years (*Mohammed, Larijani & Esmaillzadeh, 2019*), we are not aware of similar studies on schoolchildren. However, several factors can contribute to the varied prevalence of anemia and stunting at both the individual (age, sex, nutritional deficiency, food consumption, poor hygiene, intestinal helminthic infections, parents' education) and household (wealth, family income and size, food insecurity, and access to hand-washing facilities) level (*Assefa, Mossie & Hamza, 2014*; *Getaneh et al., 2017*;

*Gutema et al., 2014*; *Mesfin, Berhane & Worku, 2015*). Furthermore, most previous studies did not consider the nested structure of school data (i.e., individuals nested within the same class and classes nested within the same school) in their analysis. Few have assessed the nutritional status of schoolchildren in rural southern Ethiopia (*Grimes et al., 2017*), even if most school-aged children live in this rural area. This paper is part of a larger study on school health problems that include the co-existence of anemia and stunting (CAS), intestinal helminthic infections (*Hailu Amare & Lindtjørn, 2020*), and skin problems in the Gedeo area of southern Ethiopia. Therefore, we aimed to assess the prevalence and risk factors of CAS at the individual, household, and school level among schoolchildren in the rural Wonago district in southern Ethiopia.

## METHODS

### Study area, design, and participants

The study was conducted in the Wonago district of the Gedeo zone in Southern Ethiopia. The district is 377 km south of Addis Ababa. The district has 17 rural and four urban *kebeles*, the smallest administrative units. In 2014, Wonago's population was estimated to be 143,989 people, with 1,014 people per square kilometer of land area. The district has 26 government health facilities (six health centers and 20 health posts), two private clinics, and two drug stores, along with more than 36,000 students in three urban and 22 rural primary schools. Most residents depend on cash crops of coffee, fruit, and *ensete* (*Ensete ventricosum*).

We conducted this cross-sectional survey from February 2017 to June 2017. The study population was schoolchildren and their parents or guardians. Students aged 7–14 years were recruited in schools and their parents or guardians were contacted by visiting their homes.

### Sample size estimation

Since this study was part of a large project aiming to identify school health problems, we considered multiple factors to calculate the sample size using OpenEpi software (*Sullivan, Dean & Soe, 2009*) based on single population proportion (*Daniel, 1999*). Assuming a 95% confidence interval (CI), the maximum sample size was calculated using proportions of different variables from previous studies (e.g., anemia [27%] (*Mesfin, Berhane & Worku, 2015*), stunting [30%], and thinness [37%] (*Mekonnen, Tadesse & Kisi, 2013*); intestinal parasites infections [27%] (*Haftu, Deyessa & Agedew, 2014*), and skin tinea infection [50%] (*Ali, Yifru & Woldeamanuel, 2009*); 5% precision, and a design effect of 2 to account for multistage sampling.

To maximize the sample size, we used a prevalence of 50%. After adding a 10% non-response rate, we calculated a sample size of 845 schoolchildren.

This study targeted rural primary school-aged children. There are 22 rural primary schools in the study district. Using a three-stage cluster sampling method, we randomly assigned 4 schools to level one, 24 classes with 2,384 children to level two. We then randomly selected 36 children from each class. When more than one child in a class was

living together in the same household, one of them was selected randomly by a lottery method. In total, we randomly recruited 864 schoolchildren.

Children aged below 7 years and those older than 14 years were excluded from this study. Children who had a physical deformity and serious illness could have been excluded from this study, but we did not observe any of these cases during the data collection process.

The household of the student's parents or guardian was identified through a local guide. We randomly selected and replaced participants who dropped out of school after the selection process with participants from the same class, sex, and age by a lottery method. The recruitment process is shown in Fig. S1.

## Outcome variables

We assessed the prevalence and risk factors of concurrent anemia and stunting (CAS) among schoolchildren. CAS among children was defined when a child was both anemic and stunted. Anemia was recorded according to the WHO guidelines for school-aged children: haemoglobin <11.5 g/dl for those aged 5–11 years and <12 g/dl for those aged 12–14 years. Anemia was estimated using the haemoglobin value adjusted for altitude (*World Health Organization (WHO), 2011*). Stunting among children was defined as height-for-age Z scores <−2 SD. Children with height-for-age Z scores ≥ −3 to <−2 SD were classified as moderately stunted, whereas those with <−3 SD were classified as severely stunted (*De Onis et al., 2007*; *World Health Organization (WHO), 2009*).

## Independent variables

The independent variables at the individual level for the children were sex, age, hygienic practices: nail trimming, hand washing with soap after latrine use, and walking barefoot; and dietary *intake*, meal habit before attending school (breakfast or lunch), reported illness in the month prior to data collection, de-worming treatment in the past 6 months, and the presence of *Ascaris lumbricoides, Trichuris trichiura*, hookworm, or head lice (pediculosis) infestations. Parent individual level variables included the educational level of the mother and father.

Household factors included the wealth index, which was constructed using principal component analysis (*Vyas & Kumaranayake, 2006*) based on household assets and included electricity, radio, television, mobile phone, table, chair, bed, separate room for kitchen, cooking place, land ownership, bank account, toilet facility, floor, and roof materials; family size; the level of food insecurity; using clean drinking water; and household access to food aid in the past 6 months. School factor included participation in the school meal programme.

## Data collection tools and procedures

Ten trained enumerators conducted the interviews using a pretested and structured questionnaire that was adapted and developed in English from previous published literature (*Alelign, Degarege & Erko, 2015*; *Coates, Swindale & Bilinsky, 2007*; *Gebreyesus et al., 2015*; *Haftu, Deyessa & Agedew, 2014*) and then translated into the local *Gedeooffa* language. Children were interviewed at their schools and parents at their homes.

Information related to the child's personal hygiene was collected from the children. Measurements, such as weight, height, and haemoglobin, were taken from children in the school. To measure weight, a digital portable scale (Seca 877, Seca GmbH, Germany) was calibrated to the nearest 0.1 kg. Children were weighed in light clothing and no shoes. To measure height, a measuring board (Seca 213, Seca GmbH, Germany) was calibrated to the nearest 0.1 cm. Children were measured while standing barefoot with parallel feet, heels, buttocks, and shoulders, with their heads held upright, the backs of their heads touching the measuring board, and their hands hanging by their sides. Capillary blood samples were taken for haemoglobin measurement by trained and experienced laboratory technicians using a HemoCue Analyser Hb 301 (Angelholm, Sweden).

Stool samples were collected, processed, and examined using standard procedures (*Montresor et al., 1998*; *WHO, 1991*). Samples were collected at school in the early morning, and stored in stool cups labelled with an identification code, name, sex, age, and date. The specimens were transported in a cold-box to Dilla University Teaching and Referral Hospital, where Kato-Katz and formalin-ether concentration (FEC) techniques were used to identify helminthic infections (*Montresor et al., 1998*; *WHO, 1991*). The result of each helminthic species from two diagnostic techniques was recorded separately. We also examined the child's hair for external parasites, such as head lice.

Household information was collected from the children's parents or guardians during home visits through interviews and observations of the housing conditions. Information related to socio-demographic characteristics, household assets, household food insecurity, use of treated drinking water, access to food aid, child dietary intake, and illness in the past month were collected from the parents or guardians at each household. Information related to the habit of taking a meal regularly before school (breakfast if the child attended school in the morning and lunch if the child attended classes in the afternoon) and the reason for not eating, was also collected from the parents. Information related to de-worming treatment was collected from parents by showing them the de-worming tablet and asking, "In the past 6 months, was your child given any de-worming treatment?"

Dietary intake was measured using a 24-hour dietary recall method (*Swindale & Bilinsky, 2006*). The children's parents were asked to list the food items that the children consumed within the past 24 h, and the children were asked if they ate outside the home the *day before the survey*. We calculated the child Dietary Diversity Score (DDS) using the following nine food groups: cereals, roots, or tubers; vitamin A-rich plant foods; other fruits; other vegetables; meat, poultry, fish; eggs; pulses or legumes; milk and milk products; and food cooked in oils or fats. The food items were classified based on FAO definition (*Kennedy, Ballard & Dop, 2011*). A value of 1 was recorded if any of the disaggregated food group items were consumed, and 0 if none of the disaggregated food group items were consumed (*Swindale & Bilinsky, 2006*). DDSs were computed by summing the previously mentioned food groups, with the score ranging from 0 to 9, and the mean DDS calculated for each child.

We assessed household food insecurity based on a 4-week recall period using nine items of the Household Food Insecurity Access Scale (HFIAS) (*Coates, Swindale & Bilinsky, 2007*; *Gebreyesus et al., 2015*): (Q1) worrying about food, (Q2) unable to eat preferred foods,

(Q3) eating a limited variety of foods, (Q4) eating unwanted foods, (Q5) eating small meals, (Q6) eating fewer meals in a day, (Q7) having no food of any kind in the household, (Q8) going to sleep at night hungry, (Q9) going a whole day and night without eating anything. 'No' responses were coded as 0 and 'Yes' responses as 1. Responses for frequency of occurrence items are coded as 1 for 'rarely', 2 for 'sometimes', 3 for 'often'. Households with no affirmative responses or 1 for Q1 and 'No' responses for the other items were considered food secure, whereas those reporting 2 or 3 for Q1 or 1, 2, or 3 for Q2 or 1 for Q3 or Q4 and no affirmative responses for Q5 to Q9 were considered mildly food insecure; 2 or 3 for Q3 or Q4 or 1 or 2 for item Q5 or Q6 and 0 for Q7 to Q9 were considered as moderately food insecure; 3 for Q5 or Q6 or 1, 2, or 3 for Q7 to Q9 were categorized as severely food insecure. The score was computed by summing the responses for the frequency of occurrence for the nine items (range 0 to 27).

## Data quality control

Training was provided for all personnel participating in *data collection*, supervision, and data entry. To minimize potential bias and validate the measurement tools prior to data collection, a pre-test was conducted on 42 primary school-aged children in other schools not selected for this study. Hence, we conducted a reliability test to observe inter- and intra-observer variations and ensure reproducibility for anthropometric measurements. Observers measured the weights and heights of 10 children twice. The intra-class correlation coefficient was used to estimate the reliability of the continuous measurements (*McHugh, 2012*) (intra-observer reliability for weight: 0.94 [95% CI 0.78, 0.98], and height: 0.94 [95% CI 0.78, 0.98]; inter-observer reliability for weight: 0.92 [95% CI 0.81, 0.97], and height: 0.91 [95% CI 0.78, 0.97]). More than 90% of the intra- and inter-observer measurements show good reliability (*Koo & Li, 2016*).

To record an accurate age for the children, we cross-checked information from the school registry with the age information provided by the children's parents. The supervisors checked data for completeness and consistency.

## Statistical methods

Data were entered into a database using the double-entry system in Epi-data version 3.1 (EpiData, Odense Denmark, 2004). Inconsistencies were cleaned and missing values were addressed before analysis. After validation, the data were exported to SPSS version 20 (IBM Corp; 2011) and STATA 15 software (StataCorp LLC, College Station, TX, USA; 2017) for analysis.

The primary focus of this paper is to identify risk factors for CAS among school children. However, earlier papers have shown that CAS share risk factors for stunting and anemia among younger children. To assess this possibility, we provide separate risk estimates for anemia and for stunting as Supplemental Information. We have summarised the shared risk factors for CAS, anemia, and stunting at the end of the results section.

Descriptive statistics, frequency, percentage, mean, range, and standard deviation (SD) were calculated to describe relevant variables. The proportions were determined for categorical variables in relation to CAS. We used WHO AnthroPlus 1.0.4 software to

calculate height-for-age, weight-for-age, and body mass index-for-age Z scores according to the standard reference for children aged 5–19 years (*World Health Organization (WHO), 2009*). The WHO AnthroPlus software allows Z scores for weight-for-age to be calculated only for children up to 10 years of age (*De Onis et al., 2019*; *World Health Organization (WHO), 2009*). Therefore, children aged >10 years were excluded from the estimation of underweight prevalence.

Household asset variables were categorized and coded as either 0 for absent or 1 for present. A wealth index was then constructed using principal component analysis (*Vyas & Kumaranayake, 2006*). The internal consistency of the 14 variables was determined using a Cronbach alpha of 0.78 and Kaiser-Meyer-Olkin sampling adequacy measure of 0.8. Socioeconomic indicators (poor, middle-class, and rich) were categorized based on the first component explaining 28.3% of the data variance, with an Eigen value of 4.1.

We used three data hierarchies: school level, class level, and individual level (child or parent). Students were clustered within the same class, and classes were nested within schools. We included school- and class-level data during the analysis, and assessed potential confounding and effect modifications using multivariate, multilevel regression and stratified analysis. 'Washing hands with soap after latrine use sometimes or not always and never' were all categorized under 'not washing hands always'. 'Walking barefoot always and sometimes' were all categorized under 'walking barefoot always'. Children with no CAS and yes CAS were stratified by the categories of T. trichiura and not washing hands always.

Prior to the multivariate regression, we checked the collinearity among independent variables. Both a bivariate logistic regression without considering random effects, and a multilevel logistic regression model with random school and class effects were applied for assessment of *potential risk factors.*

Using a multivariate, multilevel logistic regression model, we identified individual-, household-, and school-level factors that may contribute to CAS. Six separate models were constructed for CAS. Model I (empty) had no covariate indicating whether to consider the random-effect model. Independent variables with $P < 0.25$ in the bivariate multilevel logistic regression model were introduced concurrently in each model. The variables with $P < 0.25$ in each independent model were also maintained in the final model. Covariate variables; sex and wealth were with $P > 0.25$ in the bivariate model were retained in the final model to control for confounding. The final model for CAS included individual child (sex, age, hand washing with soap after latrine use, walking barefoot, *T. trichiura*, and head lice), parent (mother's education), household (wealth status, family size, using treated water for drinking, access to food aid in the past 6 months), and school (participation in school meal programme) factors.

Results were calculated as crude odds ratios (cORs) and adjusted odds ratios (aORs) with 95% CIs. Independent variables with $P \leq 0.05$ in the final model were considered significant. Model fitness was checked using $-2$ log likelihood (deviance) and the Akaike information criterion (AIC). The model with the lowest deviance and AIC value was used as the final model (*Twisk, 2006*). A receiver operating characteristic (ROC) curve or area under the curve (AUC) was used to compare simple regression models with models

considering a random effect. An AUC ≥ 0.7 was considered acceptable (*Merlo et al., 2016*). In this study, the AUC for the CAS model considering a random effect was >0.80, whereas the AUC with a simple regression model was <0.7, suggesting that the *model with a random effects is better than* the model with a simple regression model for this outcome. *Homogeneity* within clusters at the school- and class-level was measured using the intra-cluster correlation coefficient (ICC).

## Ethics statement

The institutional review board of Hawassa (IRB/005/09) and Regional Ethical Committee of Western Norway (2016/1900/REK vest) provided ethical clearance. The Gedeo Zone Health Department (2/572/110) and District Education Office provided a letter of permission (435). School directors and teachers participated in discussions. We obtained informed verbal (thumb print), and written (signed) consent from study participants' parents or guardians and permission (assent) from children aged 12 years and older before the interviews. The interviews and measurements were conducted in a private place for each participant. Confidentiality of the participant's information was maintained. Children diagnosed as anemic and who tested positive for intestinal helminthic infections were referred to the nearest health institution for treatment according to standard guidelines (*Drug Administration and Control Authority (DACA) of Ethiopia, 2010*).

## RESULTS

### Background information of study subjects

The 861 schoolchildren (483 boys and 378 girls) in this study were aged 7 to 14 years, with mean [SD] age of 11.4 [1.9] years. All of the recruited children's parents or guardians participated in the household survey. The majority (89.2%; $n = 768$) of children lived with their biological parents. The father's average age was 41 years and the mother's average age was 34 years. Among heads of households, 91.4% (787/861) were men, and 8.6% (74/861) were women. The family size ranged from 3 to 14 persons (mean 6.7). Among parents, 88.4% (761/861) of mothers and 48.8% (420/861) of fathers had never attended school. More than half of the mothers (54.0%; $n = 463$) were housewives, and most fathers (77.4%; $n = 619$) were farmers. Among the participants, 33.3% ($n = 287$) lived in poor households. Table S1 summarizes the demographic and socioeconomic statuses of the schoolchildren and their parents.

### Dietary habits and food insecurity

The day before the survey, cereals, roots, or tubers (100%; $n = 861$) were consumed by all children; vitamin A-rich plant foods by 9.9% ($n = 85$); other fruits by 18.6% ($n = 160$); other vegetables by 22.6% ($n = 195$); meat or fish by 4% ($n = 35$); eggs by 17% ($n = 147$); pulses or legumes by 46.9% ($n = 404$); milk and milk products by 15% ($n = 129$); and food cooked in oil or fat by 3.7% ($n = 32$). Coffee or tea was also consumed by 87% ($n = 749$) of children a day prior to the survey. The median and interquartile range of DDS was 2 (1–3), with a range of 1 to 8 food groups consumed. The food items comprising each food group are found in the Supplementary Information (Table S2).

The median and interquartile range of HFIAS score was 2 (0–6), and scores ranged from 0 to 19. Among the 861 children, 50.7% ($n = 437$; 245 [56.1%] boys, 192 [43.9%] girls) lived in food-insecure households; 29% ($n = 253$) in mildly food insecure households, 18% ($n = 155$) in moderately food insecure households, and 3.5% ($n = 30$) in severely food insecure households. The household food-insecurity item responses are presented in the Supplementary Information (Table S3).

## School sanitation and hygiene facilities

All schools had no access to drinking water and handwashing facilities. In all schools there were pit latrines covered with cement. However, we observed that there were defections on the latrine floors, and no toilet paper were available in the toilets room.

## Prevalence and risk factors for CAS

The prevalence of CAS was 10.5% ($n = 85/810$ [95% CI 8.4, 12.6] among schoolchildren; 9.6%, $n = 44$ boys and 11.6%, $n = 41$ girls (Table 1, Tables S4 and S5). The results of the bivariate analysis for CAS are provided in Table S6.

We observed a clustering effect at the school- and class-level for CAS prevalence. The calculated ICC indicated that 6.8% of the variability in CAS among children was attributable to school-level factors and 19% to class-level factors (Table S7).

All significant variables in the bivariate were significant in the multivariate model (Tables S6 and Table S7). As shown in Table 1 and Table S7, in the multivariate analysis, the odds of CAS increased with increasing age [aOR] 1.39 [95% confidence interval 1.13, 1.71, $P = 0.002$]). Children who always did not wash their hands with soap after use of latrine (aOR 4.30 [1.21, 15.3, $P = 0.02$]), children who always walked barefoot (aOR 10.4 [2.77, 39.1, $P = 0.001$]), children infected with *T. trichiura* (aOR 1.74 [1.05, 2.88, $P = 0.03$]), or had head lice infestation (aOR 1.71 [1.01, 2.92, $P = 0.04$]) had higher CAS rates. Children using treated drinking water at their houses (aOR 0.32 [95% CI 0.11, 0.97, $P = 0.04$]) were less likely to be affected by CAS. However, we did not find significant differences between CAS and sex, nail trimming, infection with *A. lumbricoides*, mother's education, wealth, family size, access to food aid in the past 6 months, and participation in the school meal programme (Table S7).

The crude and adjusted odds ratio estimates for variables, such as hand-washing habit, and walking barefoot, were different. The association between CAS and washing hands with soap after latrine use was confounded by *T. trichiura* infection (Table 1). In addition, the association between CAS and walking barefoot was also confounded by hand washing with soap after latrine use (Table 1).

## Prevalence of anemia and stunting

Anemia alone occurred among 29.6% ($n = 240/810$) and stunting occurred in 32.3% ($n = 278/861$) of children (Tables S4 and Tables S5). Commonly shared risk factors for stunting and CAS include age, not always washing hands with soap after use of latrine, and head lice infestation. Using treated water for drinking was also found to have a protective factor against stunting and CAS (Tables S8, S9 and S10).

Peerj

**Table 1** **Bivariate and multivariate, multilevel, mixed-effect, regression analysis of CAS among schoolchildren in the Wonago district of southern Ethiopia, 2017.**

| Variables | | CAS | | Crude odds ratio (COR) (95% CI) | *P*-value | Adjusted OR (95% CI) | *P*-value |
|---|---|---|---|---|---|---|---|
| | | Yes (%) | No (%) | | | | |
| **Individual child factors** | | | | | | | |
| Sex | Boys | 44 (9.6) | 412 (90.4) | 0.81 (0.50, 1.29) | 0.37 | 0.84 (0.51, 1.39) | 0.499 |
| | Girls | 41 (11.6) | 313 (88.4) | 1.0 | | 1.0 | |
| Age in years (continuous) | Mean (SD) | | | 1.36 (1.13, 1.65) | 0.001 | 1.39 (1.13, 1.71) | 0.002 |
| Hand washing with soap after use of latrine | Always | 6 (6.2) | 91 (93.8) | 1.0 | | 1.0 | |
| | Sometimes or not always | 45 (9.6) | 422 (90.4) | 1.74 (0.68, 4.44) | 0.24 | 4.30 (1.21, 15.3) | 0.02 |
| | Never | 34 (13.8) | 212 (86.2) | 1.67 (0.58, 4.81) | 0.34 | 3.10 (0.82, 11.8) | 0.09 |
| Walking barefoot | Always | 6 (30.0) | 14 (70.0) | 5.36 (1.74, 16.5) | 0.003 | 10.4 (2.77, 39.1) | 0.001 |
| | Sometimes | 40 (10.5) | 341 (89.5) | 1.16 (0.70, 1.91) | 0.57 | 1.18 (0.68, 2.05) | 0.55 |
| | Never | 39 (9.5) | 370 (90.5) | 1.0 | | 1.0 | |
| Head lice | Yes | 42 (13.3) | 273 (86.7) | 1.69 (1.05, 2.74) | 0.03 | 1.71 (1.01, 2.92) | 0.04 |
| | No | 43 (8.7) | 452 (91.3) | 1.0 | | 1.0 | |
| *T. trichiura* | No | 41 (8.8) | 424 (91.2) | 1.0 | | 1.0 | |
| | Yes | 42 (12.4) | 298 (87.6) | 1.59 (0.99, 2.55) | 0.05 | 1.74 (1.05, 2.88) | 0.03 |
| **Household factors** | | | | | | | |
| Using treated drinking water | Yes | 4 (4.1) | 94 (95.9) | 0.28 (0.10, 0.80) | 0.01 | 0.32 (0.11, 0.97) | 0.04 |
| | No | 81 (11.4) | 631 (88.6) | 1.0 | | 1.0 | |
| **School factor** | | | | | | | |
| Participates in school meal programme | No | 55 (13.7) | 345 (86.3) | 1.0 | | 1.0 | |
| | Yes | 30 (7.3) | 380 (92.7) | 0.48 (0.22, 1.03) | 0.06 | 0.29 (0.07, 1.17) | 0.08 |

**Notes.**
CAS, concurrent anemia and stunting; CI, confidence interval; OR, odds ratio

## DISCUSSION

Based on a standard definition, CAS was found to be a moderate public health problem among schoolchildren in southern Ethiopia (*De Benoist et al., 2008*; *De Onis et al., 2019*). Increasing age, not always washing hands with soap after using latrine, walking barefoot, *T. trichiura* infection, and head lice infestation were found to be associated with CAS. Using treated water for drinking was found to have a protective effect against CAS. Stunting and CAS share some of the same risk factors, including age, not washing hands with soap after use of latrine, and head lice infestation. Using treated drinking water was also a common shared protective factor for stunting and CAS. Significant clustering effect was observed at the school- and class-level for CAS prevalence.

The rate of CAS was 10.5% among schoolchildren. This rate is lower than the recent report of 23.9% for Ethiopia (*Mohammed, Larijani & Esmaillzadeh, 2019*), 21.5% for India, and 30.4% for Peru (*Gosdin et al., 2018*) among children under the age of 5 years. These variations could be due to different study settings and participant ages. Moreover, unlike most studies, we conducted our study in rural areas, which could also have influenced prevalence rates.

Using age as a continuous variable, we observed significantly increased odds of CAS with increasing age, as described previously in Ethiopia (*Alelign, Degarege & Erko, 2015*; *Bogale et al., 2018*; *Tariku et al., 2018*). This could be explained by the older school-age children are in the transition period to puberty, which increases nutritional demands (*Soliman, De Sanctis & Elalaily, 2014*).

Our results revealed an increased risk of CAS among children who always not washed their hands with soap after using the latrine compared to children who always washed their hands. This finding is supported by previous studies (*Gosdin et al., 2018*; *Mahmud et al., 2015*). Good hygiene is important to avoid acquiring infections and, thus, avoid CAS caused by helminthic infections. Moreover, walking barefoot may partly be indicative of low socio-economic status and poor hygiene. Relative to children who never walk barefoot, children who always walk barefoot had 5-times the risk of CAS in the bivariate analysis; introducing the habit of hand washing in the multivariate analysis, the risk of CAS was 10-times higher, potentially indicating the confounding effect of hand washing. Furthermore, these findings should be interpreted with caution as the number of children who always washed hands with soap after the use of latrine and walked barefoot were few in this study.

Species-specific analysis of intestinal helminthic infections revealed that *T. trichiura* is significantly associated with CAS. This finding was also consistent with the previous findings (*De Gier et al., 2016*; *Shang et al., 2010*; *Stephenson, Holland & Cooper, 2000*). The link between intestinal helminthic infections and CAS could be due in part to the inflammatory reactions to worm infection in the intestinal mucosa, resulting in a loss of appetite, reduced food intake, and impaired iron absorption (*Shang et al., 2010*; *Stephenson, Holland & Cooper, 2000*). No significant association was found between CAS and hookworm infections, probably due to low statistical power because the number of children with hookworm infection was low.

The significant effect of head lice infestation on risk of CAS may be associated with low socio-economic status and poor hygiene (*Moosazadeh et al., 2015*). We found lower odds of CAS among children using treated drinking water, suggesting a protective effect against intestinal helminthic infections (*Matangila et al., 2014*; *Strunz et al., 2014*; *Worrell et al., 2016*). Wealth status was also associated with CAS among children (*Mohammed, Larijani & Esmaillzadeh, 2019*), but we found no difference in any of these odds across poor, middle-class, and rich households, findings in agreement with *Alelign, Degarege & Erko (2015)*. There could be other socio-economic factors not investigated in this study.

In this study, neither dietary diversity nor household food security was associated with CAS. The lack of association may be due to similarities in the dietary diversity score and mild food insecurity in the area. A study from Kenya reported that participation in a school meal programme lowers the risk of anemia, and stunting (*Neervoort et al., 2013*). Consistent with the previous finding in southern Ethiopia (*Shaka & Wondimagegne, 2018*), our study did not find a significant difference in the rates of CAS among children participating in a school meal programme and those children who did not participate, although the observed association could be described as borderline significant ($P = 0.08$). Only cereals were served at school; therefore, the food being served to the children at school was deficient in micronutrients and may explain why we failed to detect a significant difference.

The strengths of this study include: we used a large representative sample of schoolchildren and applied a multilevel, mixed-effect model to identify risk factors for CAS. We also reported the risk factors for anemia, and stunting. We diagnosed helminthic infections using two standard techniques, the Kato-Katz and formalin-ether concentration. Data were clustered. Thus, individual within-school and within-class dependencies (similarities) were identified and measured using intra-cluster correlation coefficient.

Our study has some limitations. First, we used a cross-sectional design; thus, causality between the outcome and independent variables cannot be determined with certainty. Second, due to homogeneity issues, most school factors in this study (e.g., sanitation and hygiene facilities at the schools) were not modelled. Third, we did not measure serum ferritin levels to assess iron status, and we did not assess the presence of malaria. Recording dietary information for consecutive days and measuring the quantity of food consumed could have improved the dietary information. Unfortunately, we collected dietary information using only 24-hour dietary recall, and we did not measure the quantity of the food. Further study in the area should assess the micronutrient levels in schoolchildren. Our findings related to food insecurity could also be affected by over- or under-reporting on the HFIAS questions. Over-reporting could arise if responses were related to intentions to get food aid. Under-reporting of the last three HFIAS questions could also occur due to fear of disclosing severe food insecurity related to cultural perceptions (*Kabalo et al., 2019*). The results of hand washing may be affected by self-reported bias. Children who had their head shaved could have been affected by head lice, but we did not observe any with these characteristics during the examination. However, head lice could have been under-estimated. Furthermore, including only children at rural schools may affect the external validity of this study. However, we think that most school-aged children attended

school. Thus, the findings can be generalized to rural school-aged children in the same region where culture and living standards are similar.

## CONCLUSIONS

Our findings suggest the need for more effective nutrition programmes to tackle CAS among schoolchildren in rural areas of Ethiopia. Interventions that improve hygiene can reduce the morbidity caused by intestinal helminthic infections. Provision of safe drinking water and promotion of treated water could also reduce comorbidities in schoolchildren. School teachers should work with health workers to provide health education about personal hygiene protection. Therefore, individual-, household-, and school-level intervention activities must be integrated to improve the health of schoolchildren.

## ACKNOWLEDGEMENTS

We are grateful to the schoolchildren, parents, and guardians who participated in this study. We also thank the data collectors, supervisors, Gedeo Zone Health Department, Wonago District Education Office, school directors, and teachers. We are also grateful to the Ethiopian Public Health Institute for providing the Kato-Katz template. We would like to thank Dilla University Teaching and Referral Hospital for providing us with a laboratory examination room.

### Funding

Norwegian Programme for Capacity Development in Higher Education and Research for Development/South Ethiopia Network of Universities in Public Health (NORHED/SENUPH) project provided funding in the form of a grant awarded to Bernt Lindtjorn (ETH-13/0025). The funders had no role in study design, data collection and analysis, decision to publish, or preparation of the manuscript.

### Grant Disclosures

The following grant information was disclosed by the authors:
Norwegian Programme for Capacity Development in Higher Education and Research.
Development/South Ethiopia Network of Universities in Public Health (NORHED/SENUPH).
Bernt Lindtjorn: ETH-13/0025.

### Competing Interests

The authors declare there are no competing interests.

### Author Contributions

- Hiwot Hailu Amare and Bernt Lindtjorn conceived and designed the experiments, performed the experiments, analyzed the data, prepared figures and/or tables, authored or reviewed drafts of the paper, analysis tools, and approved the final draft.

## Human Ethics

The following information was supplied relating to ethical approvals (i.e., approving body and any reference numbers):

The institutional review board of Hawassa (IRB/005/09) and Regional Ethical Committee of Western Norway (2016/1900/REK vest) provided ethical clearance. The Gedeo Zone Health Department (2/572/110) and District Education Office provided a letter of permission (435/ወ/ወ/ኢስ/ማ). School directors and teachers participated in discussions. We obtained informed verbal (thumb print), and written (signed) consent from study participants' parents or guardians and permission (assent) from children aged 12 years and older before the interviews. The interviews and measurements were conducted in a private place for each participant. Confidentiality of the participant's information was maintained. Children diagnosed as anemic and who tested positive for intestinal helminthic infections were referred to the nearest health institution for treatment according to standard guidelines (*Drug Administration and Control Authority, DACA)of Ethiopia(2010*).

## Field Study Permissions

The following information was supplied relating to field study approvals (i.e., approving body and any reference numbers):

The Gedeo Zone Health Department (2/572/110) and District Education Office (435/ወ/ወ/ኢስ/ማ) provided letters of permission.

## Data Availability

Data are available in the Supplemental Files.

## Supplemental Information

Supplemental information for this article can be found online at http://dx.doi.org/10.7717/peerj.11158#supplemental-information.

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
