# Peer review of "Concurrent anemia and stunting among schoolchildren in Wonago district in southern Ethiopia: a cross-sectional multilevel analysis"

_PeerJ, doi:10.7717/peerj.11158_

## Round 0.1 · original submission · Major Revisions

The sample size was estimated with a 50 percent prevalence assumption to maximize sample size. However, the individual prevalence of anemia and stunting is much lower, and concurrent was expected to be lower than individual. Please clarify this.

Results
While describing dietary diversity and food insecurity, 95% CIs are used. Here first its variability is mentioned and use of standard deviation instead of 95% CI is suggested. Indeed, the median and interquartile range would be better measures considering the discrete and ordinal nature of the data. When generalizing it is better to quote 95% CI.

Statements about associations between CAS, hand washing, Trichuria infection, and barefoot walking are not clear. Please use a diagram to show these associations and interrelationships.

Please look at the effects of handwashing. Its confidence intervals are wide, and there are very few observations in the “Always” group which is a referent group. Comment about issues in generalizing these effects.

Discussion

Please avoid the use of the words “predictors” and “determinants” considering the cross-sectional nature of the study. Instead, write “associated”.

In a statement, Line 358-359- write “not washing hands with soap after latrine”. Correct this wherever it is presented as a risk factor. not washing hands with soap is a risk factor while washing hands offers protection. The manuscript is talking about risk factor. Please carefully look wherever this word is used.

As suggested earlier, a summary diagram showing interrelationships among variables will help readers understanding the results.

·

Basic reporting

Introduction
Good; but in text reference of the document is not appropriate. Which reference methods do you used?
For example line 24, line 28&29… “stunting among children aged under-5 years (Mohammed, Larijani & Esmaillzadeh, 2019), … and access to hand-washing facilities) level (Assefa, Mossie & Hamza, 2014; Getaneh et al., 2017; Gutema et al., 2014; Mesfin, Berhane & Worku, 2015)”. I think your reference style is Harvard method and in Harvard reference method, If the authors greater than or equal to three, in text reference written as Getaneh et al.,2017. So you have to correct accordingly the whole in text reference

Methodology
In study area, design, and participants
You wrote “ … and 4 urban kebeles... 6 health centers …. , 2 private clinics, and 2 drug stores, … 3 urban and 22 rural primary schools.” All number below 10 have to be written in letters like, four urban kebeles … six health centers…..
In sample size estimation
-Were the sample sizes proportionate or disproportionate from each class? And your calculated sample size was 845 schoolchildren but you recruited 864 schoolchildren why? If so, why it is important to calculate sample size? I mean why you are not recruited the sample proportionally?
- You said “We replaced participants who dropped out of school after the selection process with participants of the same class, sex, and age.” Please rewrite this statement including how you had replaced. I mean through which sampling method?
- …… sampling (Ali, Yifru & Woldeamanuel, 2009; Haftu, Deyessa & Agedew, 2014; Komba & Mgonda, 2010; Mekonnen, Tadesse & Kisi, 2013; Mesfin, Berhane & Worku, 2015). In text reference should be improved.

In data collection tools and procedures
Ten trained enumerators conducted the interviews using a pretested, structured questionnaire that was adapted and developed in English …. From where adapted these questionnaire? And you have to provide the references that you have adapted.
In statistical method
You wrote; “Independent variables with P < 0.05 in the final model were considered significant”. Write as … with P ≤ 0.05 …
In ethics statement
You stated that, “Participant privacy and confidentiality were maintained”.
How participants’ privacy and confidentiality were maintained? Please state in what means that you maintain the privacy and confidentiality of participants.
In results and discussion
Well written, but similar to introduction and sample size estimation the in text reference needs improvement like:-
… for Ethiopia (Mohammed, Larijani & Esmaillzadeh, 2019)
… as described previously in Ethiopia (Alelign, Degarege & Erko, 2015 …
… nutritional demands (Soliman, De Sanctis & Elalaily, 2014).
… findings (de Gier et al., 2016; Shang et al., 2010; Stephenson, Holland & Cooper, 2000).
Revision of the above listed comments of the manuscript is needed. The topic of this manuscript falls within the scope of journal of PeerJ .

Experimental design

No comment

Validity of the findings

No comment

Additional comments

Check the whole manuscript and ensure the proper use of words, grammar ,capital and small letters
For example - in abstract method parts you wrote … helminth infections … to say … helminthic infections
- in introduction line 23 you wrote “anaema” in terms of anemia
- in Study area, design, and participants line 5 you wrote … “health centres” ….. to say health centers and line 10 you wrote … “parents or guardian were contacted”… to say parents or guardians were
The document needs eligible criteria and I never seen it in the document. I mean the manuscript states measurement of height and age for assessing stunting but we cannot assess stunting in the disabled children like anatomical deformed ones, because the stature does not show the nutritional problems.
Does this journal not require showing details on Disclosure about funds, Conflicts of Interest to the publication of this manuscript and Authors’ Contributions? However, my opinion suffices to just show in the main documents.

Reviewer 2 ·

Basic reporting

The text in English is easy to read. No major issue.
A little more could be added about the justification of undertaking the study in the targeted region.
At some places, some newer references are needed to be added regarding stunting and anemia in Africa.
The structure is fine. The methods are well described.
The results are relevant and bringing a new knowledge.

A few minor comments:

Abstract
Line 28-29: Could be good to indicate what was the sampling strategy. Random selection of 4/22 schools in the district and 24 classes from those 4 schools. May be good to indicate precisely that it was a 3-stage sampling.

Line 44: Is it a positive association for all listed factors?

Introduction
Line 58: Would be good to have a newer reference that this old one. New references about stunting and/or anemia in some countries in Africa can be found on PubMed.

Line 68-69: Same observation. Would be better to add some newer references.

Line 82: Why this region? What is its vulnerability or special status in Ethiopia regarding the health problems targeted by the larger study and by the outcomes covered by this paper?

Methods
Line 104-105: May be good to indicate from which reference is precisely coming each of these percentages. In the Introduction or the Study area, some already mentions of the situation in the area would make easier the choice of the percentage for the sample size calculation.

Discussion
Line 357: What criteria justifies a qualification of "moderate", "high" or else and what are the references?

Experimental design

Well done.
Methods described with sufficient details.
Good statistical analyses performed.

Validity of the findings

Results are strong and increasing the knowledge on a very complex dual burden.

Additional comments

An interesting paper exploring a complex public health issue.
The Abstract could better highlight the 3-stage sampling strategy.
And a number of additional newer studies on anemia and/or on stunting in Africa could be easily found with a literature on Pubmed or other well known public health journals.

---

## Round 0.2 · Minor Revisions

The manuscript is now revised and is in good shape, however a few minor corrections are still needed.

Figure-2 added in revision doesn't clarify interrelationships as suggested and may be deleted.

Please respond to the comments of the Reviewer.

·

Basic reporting

The manuscript has been improved significantly; however there are some minor issues should be solved before final publication

1. The study conducted in Wonago district only, so adding the phrase Wonago district to the title will make it more specific and improve the manuscript title.
2. In some area of your document there is un clear statement, like;
In line169; Independent variables included at school level were participation in a school meal programme. “Were” is not correct in this context, do you mean where? Please rewrite the statement
3. There is inconsistency in writing words; example , sometimes written as anaemia, sometimes as anemia and in other areas of your document as anema. Please write them in a consistent way and correctly.
4. As I commented before, still there are so many spilling and grammar errors in your document please take it in to consideration and correct the whole documents.
Exampls;
Line 45... southern Ethiopia and vary across classes and schools. “Vary” change in to varies
Line 59… anaemic or stunted… “anaemic” change in to anemic
Line 61… school access and enrolment has improved,…” has “ change in to have
Line 68 …. few recent global report….. “ report” change in to reports
Line 74 …. with a prevalence rates… “a prevalence rates” change in to prevalence rates/ a prevalence rate
Line 85…co-morbidity of anema and stunting… “anema” change in to anemia
Line 97…. problems that includes… “includes” change in to include
Line 108…”kilometre”…. Change in to kilometer
Line 252 &253… “missing values addressed before analysis”. Change in to missing values were addressed…
Line 275 …”indictor” … change in to indicator
Line 295… ‘in each models’ … change in to each model
Line 361…”in all school”… change in to ... in all schools
Line 398 … common shared … changed in to commonly shared
Line 414 The rates of CAS was… change in to the rate of CAS was…
Line 463 …only cereals was served…. “Was” change in to were
Line 470 … two standard technique…” technique” change in to techniques
Line 501 “Acknowledgements” change in to Acknowledgments

Experimental design

The study is done using appropriate and very correct and impeccably used methodologies. Good statistical analyses was performed.

Validity of the findings

Results are strong and increasing the knowledge on a very complex dual burden.
The conclusions are relevant and important, supported by the generated data.
Even though it was made in Ethiopia, in a particular district setting, the research question and methodologies can, and should, be used in other contexts.

Additional comments

A good paper to Visiting a multifaceted public health problems.
I consider that, it is a very important paper and recommend it for publication after corrected this minor comments
The topic of this manuscript falls within the scope of Journal of PeerJ

---

## Round 0.3 · accepted · Accept

The manuscript is now revised as suggested. It is recommended for publication.